# Trend and epidemiological patterns of animal bites in Golestan province (Northern Iran) between 2017 and 2020

Vahid Bay[1©], Mehdi Jafari[1,2], Mohammad Reza Shirzadi[3], Ali Bagheri[4], Irvan Masoudi Asl[1©]*

1 Department of Healthcare Services Management, School of Health Management & Information Sciences, Iran University of Medical Sciences, Tehran, Iran, 2 Health Managers Development Institute, Ministry of Health and Medical Education, Tehran, Iran, 3 Center for Communicable Diseases Control, Ministry of Health and Medical Education, Tehran, Iran, 4 Infectious Diseases Research Centre, Golestan University of Medical Sciences, Gorgan, Golestan, Iran

© These authors contributed equally to this work.
* aramiyan3@gmail.com

**Data Availability Statement:** All relevant data are within the paper and its Supporting Information files.

## Abstract

### Background

Rabies is one of the oldest zoonosis viral diseases, which still remains as one of the most important threats to public health in the 21st century.

### Methods

This cross-sectional study examined epidemiologic features of all 33,996 cases of persons bitten by animals and referred to the rabies prophylaxis centers in Golestan province between March 2017 and March 2020. Factors included demographic information of the victim (age, gender, and occupation), type of invasive animals (dog, cat, and other types), time of bite (year, month, and hour), place of residence (urban or rural), and injury and treatment statuses. We also obtained national and provincial animal bite incidence data for all of Iran and for Golestan province for the longer interval 2013–2020 to examine broader time trends. We used SPSS version 19, QGIS version 3.1, and Excel 2013 to generate frequency distributions and descriptive statistics.

### Results

The incidence rates of animal bites in Golestan province and Iran as a whole both increased smoothly. The latest incidence rate of animal bites in Golestan was 652 per 100,000 people, almost three times the overall national figure for 2020. Most cases of animal bites (67.6%) occurred in rural areas, and 36% of the victims aged under 19 years old. Dog and cat bites accounted for the great majority of cases (89% and 8%, respectively). The highest rate of animal bites was reported in the spring (30.8%). The lower limb was the most commonly bitten area in these individuals (64.6%). Of note, 87% of the cases received incomplete prophylactic post-exposure treatment, and 18% received immunoglobulin.

**Funding:** The author(s) received no specific funding for this work.

**Competing interests:** The authors have declared that no competing interests exist.

## Conclusion

The increasing rates of animal bites in the study area as well as the higher rate compared to the national average indicates the need for further review of animal bite control programs.

## Introduction

Rabies is a zoonosis viral disease causing encephalomyelitis in humans and all warm-blooded mammals. This disease is one of the most important zoonosis and the oldest one, it's the importance of which is due to its high lethality (100%) and economic costs [1–4].

The virus is distributed worldwide with the exception of Antarctica. As a result, about 10 million people annually bitten by animals worldwide are under the treatment for rabies prophylaxis [5,6]. The virus kills approximately 59,000 people worldwide each year, the majority of which occur in the populations of Africa and Asia [1].

In Iran, rabies has long been endemic to Iranian wildlife and frequently infects domestic animals [7]. In Iran, in the last 30 years, statistics of animal bites cases has increased with the incidence rate of 35 per 100.000 population in 1987 up to 177 per 100.000 people in 2016 [5].

According to the latest incidence status of animal bites in the country, Golestan province, along with Ardabil, North Khorasan, Charmahal-Bakhtiari provinces, are known as the provinces with high rates of animal bites. Notably, the incidence rate of animal bites in this province has increased from 503 cases in 2011 up to 557 cases per 100, 000 people in 2016 [5].

Therefore, this study aimed to investigate the trend and epidemiological patterns of animal bites in Golestan province.

## Materials and methods

We conducted a cross-sectional study including all 33,996 cases of persons bitten by animals and referred to anti-rabies centers for prophylaxis in Golestan province from March 2017 to March 2020. We also compiled national and Golestan data on animal bites reported from 2013–2020 to study longer-term incidence trends.

The variables investigated in this study included demographic information of victims (age, gender, and occupation), type of invasive animals (dog, cat, and other types), time of onset of animal bite (year, month, and hour), place of animal bite (urban or rural), injury status (number of injured limb), and type of medical services provided. Golestan provincial data reflecting epidemiological factors were collected from the electronic animal bites registration system. National data on overall and province-specific incidence rates 2013–2020 were collected from the Center for Communicable Diseases Control of the Ministry of Health of Iran. After collecting the research's data, they were entered into SPSS software version 19, QGIS version 3.1, Excel 2013, and descriptive statistics were the used to analyze the data.

This study was approved by the Research Ethics Committee of Iran University of Medical Sciences under NO IR.IUMS.REC.1398.716. The patients' recorded information were kept confidential and all data was completely anonymized prior to access. This research was conducted in terms of the ethical principles and national norms and standards of conducting medical research in Iran.

## Results

This study showed that the incidence of animal bites in Golestan province has increased from 492 cases per 100,000 people in 2013 up to 652 cases in 2019–2020. Additionally, the incidence

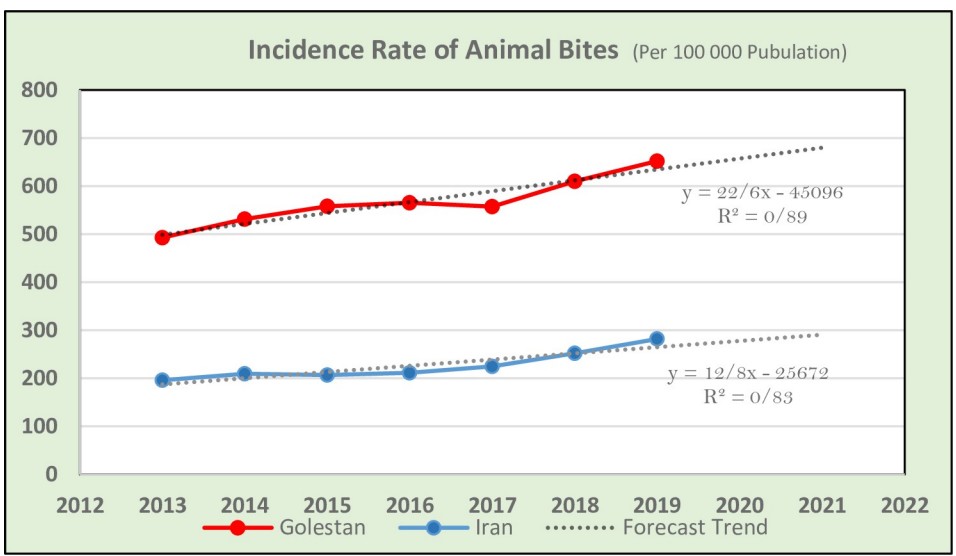

**Fig 1. Incidence trend of animal bites per 100,000 people in Golestan province and in Iran.**

of animal bites in Iran has increased from 196 to 282 in the above-mentioned period (see Fig 1). In Golestan province, Gomishan, Aliabad, and Maraveh-Tappeh counties had the highest incidence rates (934 to 957 cases per 100,000), while Minudasht County had the lowest incidence (391 cases per 100,000) in 2019–2020 (see Fig 2).

The results of this study show that 78.5% of the bitten animal individuals were men. The highest number of bites was related to the age groups of 1–9 and 10–19 years old with 18.8% and 19.2%, respectively, indicating that the rate of bites in age groups decreased with aging. In addition, 67.6% of the population were living in rural areas and 0.3% of them were non-Iranians (Table 1).

This study showed that most of the animal bites were reported in spring and summer and the least in autumn and winter. As well, the highest rate of bites was reported between 12–18

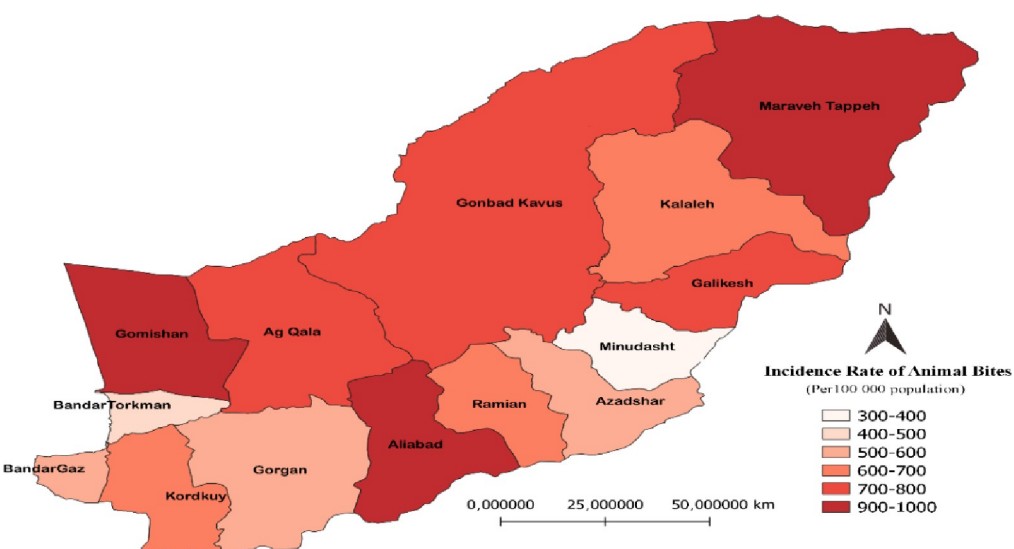

**Fig 2. Animal bite incidence rates in Golestan counties (2019–2020).**

**Table 1. Demographic information of the selected bitten animal patients.**

| variable | | Number | percent |
|---|---|---|---|
| gender | male | 26688 | 78/5 |
| | female | 7308 | 21/5 |
| Residence area | City | 11050 | 32/4 |
| | Village | 22946 | 67/6 |
| Age group | 1–9 | 6404 | 18/8 |
| | 10–19 | 6512 | 19/2 |
| | 20–29 | 5919 | 17/5 |
| | 30–39 | 5631 | 16/5 |
| | 40–49 | 3994 | 11/7 |
| | 50–59 | 2953 | 8/7 |
| | 60 years and older | 2582 | 7/5 |
| occupation | Child | 3638 | 10/7 |
| | Student | 7740 | 22/7 |
| | housewife | 4306 | 12/7 |
| | Employee | 1288 | 3/8 |
| | Freelancer (farmer and rancher, worker, etc.) | 4526 | 13/3 |
| | Other | 2670 | 7/8 |

o'clock with a rate of 37.1% and the lowest between 0–6 o'clock with a rate of 3.1% of the total bites. Temporal aspects of the animal bites are shown in Table 2.

This study also showed that most of the animal bites were related to dog and cat with a rate of 96% of total bites; 49% of these animals had owners. Additionally, 58% of animals' bites resulted in one injury and 28% of them caused two injuries. The most common bitten limbs were the hip and lower limbs with 64.6%. Also 87% of the patients received incomplete vaccine and 18% of the patients received immunoglobulin. During performing this study, one case of human rabies was reported. Information on the type of invading animal, place of the injury, and medical services provided are shown in Table 3.

## Discussion

According to the World Health Organization's report, Asia and Africa have the highest rate of animal bites [1]. In Iran, the rate of animal bites in most areas has an increasing trend [5]. We confirmed the trend up to the present: the incidence of animal bites in Iran has increased from 196 to 282 from 2013–2020. The rate of increase for animal bites in Golestan province paralleled that of Iran as a whole, although at substantially higher rates than the national rate. The

**Table 2. Temporal aspects of the animal bites.**

| variable | | Number | percent |
|---|---|---|---|
| Bite season | Spring | 10471 | 30/8 |
| | Summer | 8939 | 26/3 |
| | Fall | 7547 | 22/2 |
| | winter | 7039 | 20/7 |
| Bite hours | 0–6 | 1064 | 3 |
| | 6–12 | 9350 | 27/5 |
| | 12–18 | 12623 | 37 |
| | 18–24 | 10960 | 32/2 |

**Table 3. Information on the type of invading animal, place of the injury, and medical services provided.**

| variable | | | Number | percent |
|---|---|---|---|---|
| Invasive animal type | Dog | | 30220 | 89 |
| | Cat | | 2718 | 8 |
| | Horses, ass, camels | | 220 | 0/7 |
| | cow, goats and sheep | | 357 | 1 |
| | Mice (types of mice, guinea pig, etc.) | | 328 | 1 |
| | Reynard, fox and wolf | | 52 | 0/2 |
| | Other animals | | 101 | 0/3 |
| Place of injury | Head, neck and face | | 678 | 2 |
| | Upper limbs (shoulder, arm, forearm, hand) | | 10465 | 30/8 |
| | Chest, abdomen, back | | 878 | 2/6 |
| | Hip and lower limbs | | 21976 | 64/6 |
| Medical service provided | vaccine | complete vaccine | 4351 | 12/8 |
| | | incomplete vaccine | 29576 | 87 |
| | Immunoglobulin | | 6119 | 18 |

current study showed that the rate of animal bites in Golestan has increased from 492 cases in 2013 up to 652 cases per 100.000 people in 2019–2020. One of the reasons for the higher rate of animal bites in this province may be due to the location, which is in the center of animal husbandry and agriculture in Iran, naturally increasing the exposure of both humans and animals.

Our data showed that the most animal bites in Golestan occurred in men. Animal bites have also been reported more commonly in men in other Iranian studies [8–13]. In some countries, the rate is higher among women [14], probably reflecting economic and behavioral differences. Unlike western countries, keeping animals at home is usually avoided in Iran. Additionally, the high rate of bites in men can be related to their more presence in society due to occupational and non-occupational activities and more boldness in contact with animals.

According to the international data, most cases of human rabies have been reported in rural areas [1]. In the present study, 67% of the animal bites were reported from the rural community, which is consistent with most of the domestic data. However, in a study conducted in Abadeh, 57% of the bitten animal individuals were living in cities, which was not consistent with our study [8,9,11–13]. This issue can be due to the nature of rural occupations (husbandry and agriculture) as well as the presence of dogs in most rural households in the region.

In the present study, most of the bitten animal individuals were found to be related to the age groups of 1–9 and 10–19 years old, which included 38% of the bitten animal individuals. According to the international child data, the highest rate of animal bites by dogs belongs to children, the highest incidence rate of which is in mid to late childhood [14]. In other foreign studies conducted in India and Pakistan, cases in the age group under 15 and 17 years old were 34 and 48% of the bitten animal individuals, respectively, which was consistent with the data of our study [15,16]. However, in domestic studies, the most bites varied between the age groups of 10 to 40 years old [11], indicating the involvement of young age groups in this province, which can be due to the curiosity of this age group and provoking of animals.

According to the World Health Organization's report, 76–94% of animal bites are caused by dogs in countries with low and medium incomes [14]. In our study, the most common causes of bites were dog and cat, which accounted for 89% and 8% of the bites, respectively. In a review study conducted in Arab countries, the dog was found as the most important cause of transmitting the virus to humans, with the exception of Oman, where the fox was introduced

as the most important cause of transmitting the virus [17]. In other domestic studies, dog was also reported as the most common bite cause, which was consistent with our study [9,11–13,18]. This matter can be due to the large presence of dogs as domesticated animals in different human communities with different uses.

The present study showed that we mostly observe the highest rate of bites in the province in spring and summer, and the highest rate of bites has been reported in terms of daily hours between 12–18. In domestic studies, the highest rate of bites was mentioned to be in spring and summer and the lowest rate was reported in winter. Also, in a study conducted in Khorramshahr, the highest number of bites was reported between 12 to 18 o'clock, which its results were in line with our study [8,9,11,12,19]. The higher incidence of bites in spring and summer can be attributed to the expansion of husbandry and agricultural activities in spring and summer in this region.

In the present study, only one case of human rabies was reported in the studied geography. These data are consistent with data obtained from other parts of the country [5]. Considering the high incidence of animal bites in these areas, it can be said that the reason for the low rate of human deaths resulted from rabies is the appropriate coverage of post-exposure prophylaxis (PEP) services in the Iran's health system.

According to the findings of this study, freelancers, students, housewives, and children had the highest rate of bites with 41, 24, 13, and 11%, respectively. In the studies conducted in Najafabad and Bardsir, freelancers and workers had the highest rate of bites followed by the students, which were in line with our study [13,18]. However, in a study conducted in Mashhad, after freelancers, people with governmental jobs recorded the highest rate of bites, part of which was inconsistent with our study [12]. In our study, the increased rate in bites in this group of people can be due to having freelance jobs such as animal husbandry and agriculture with animals like dog in the village, and also in students due to their specific age and curiosity to provoke animals.

## Conclusion

Observing the increasing trend of animal bites in the study area (about three times higher than the national average) and consequent negative health, psychological, social, and economic effects caused by it, warn the need to review the current animal bites management programs based on the regional epidemiological patterns.

## Supporting information

**S1 File. Information file.** Information about: incidence rate (in cities, provinces and countries) and other epidemiological characteristics of animal bites.
(XLSX)

## Acknowledgments

We appreciate the cooperation of the Department of Zoonosis Diseases of the Ministry of Health of Iran as well as the Deputy Minister of Health of Golestan University of Medical Sciences in conducting this study.

## Author Contributions

**Data curation:** Mohammad Reza Shirzadi, Ali Bagheri.

**Methodology:** Mehdi Jafari.

**Project administration:** Vahid Bay.

**Software:** Vahid Bay.

**Writing – original draft:** Vahid Bay.

**Writing – review & editing:** Irvan Masoudi Asl.

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
