## [Decision Letter · Decision Letter 0]

12 Feb 2021

PONE-D-20-38907

Trend and epidemiological patterns of animal bites in Golestan Province (North Iran) between 2017 and 2020

PLOS ONE

Dear Dr. masoudiasl,

Thank you for submitting your manuscript to PLOS ONE. After careful consideration, we feel that it has merit but does not fully meet PLOS ONE’s publication criteria as it currently stands. Therefore, we invite you to submit a revised version of the manuscript that addresses the points raised during the review process.

In addition to addressing all reviewers' comments, especially comments made by reviewer #2 on methods used in the manuscript, this manuscript requires a good English edit.

We look forward to receiving your revised manuscript.

Kind regards,

Amir Radfar, MD,MPH,MSc,DHSc

Academic Editor

PLOS ONE

Journal Requirements:

2. Please clarify the source of the socio-demographic data that were used for this study. If prospectively obtained, please clarify the consent process that was used to collect the data. If it was included in the patient charts, please clarify that and include whether or not the data were anonymized before the authors accessed them. Please note that PLOS ONE requires that reported research meets all applicable standards for the ethics of experimentation and research integrity (http://journals.plos.org/plosone/s/criteria-for-publication#loc-6).

3. In the Methods section please provide the source of the national and provincial data.

4.  We suggest you thoroughly copyedit your manuscript for language usage, spelling, and grammar. If you do not know anyone who can help you do this, you may wish to consider employing a professional scientific editing service.  

6. We note that Figure 2 in your submission contain map images which may be copyrighted. All PLOS content is published under the Creative Commons Attribution License (CC BY 4.0), which means that the manuscript, images, and Supporting Information files will be freely available online, and any third party is permitted to access, download, copy, distribute, and use these materials in any way, even commercially, with proper attribution. For these reasons, we cannot publish previously copyrighted maps or satellite images created using proprietary data, such as Google software (Google Maps, Street View, and Earth). For more information, see our copyright guidelines: http://journals.plos.org/plosone/s/licenses-and-copyright.

6.1.    You may seek permission from the original copyright holder of Figure 2 to publish the content specifically under the CC BY 4.0 license. 

6.2.    If you are unable to obtain permission from the original copyright holder to publish these figures under the CC BY 4.0 license or if the copyright holder’s requirements are incompatible with the CC BY 4.0 license, please either i) remove the figure or ii) supply a replacement figure that complies with the CC BY 4.0 license. Please check copyright information on all replacement figures and update the figure caption with source information. If applicable, please specify in the figure caption text when a figure is similar but not identical to the original image and is therefore for illustrative purposes only.

Reviewers' comments:

Reviewer's Responses to Questions

**Comments to the Author**

1. Is the manuscript technically sound, and do the data support the conclusions?

Reviewer #1: Yes

Reviewer #2: Partly

2. Has the statistical analysis been performed appropriately and rigorously? 

Reviewer #1: I Don't Know

Reviewer #2: N/A

3. Have the authors made all data underlying the findings in their manuscript fully available?

Reviewer #1: No

Reviewer #2: Yes

4. Is the manuscript presented in an intelligible fashion and written in standard English?

Reviewer #1: Yes

Reviewer #2: Yes

5. Review Comments to the Author

Reviewer #1: Thank you for submitting this manuscript.

GENERAL

Please add page and line numbers to the document. It makes it much easier for reviewers to cite specific parts for comments and queries.

Please indent the first line of each paragraph. That will make the manuscript easier to follow.

I think that the manuscript would benefit from the services of an English-language editor. It could be more concise and clear.

METHODS

The source data from Golestan and from the entire nation should be described more clearly. I would refer to incident animal bite cases for all Iran (including Golestan) 2013-2020; and incident animal bite cases plus relevant epidemiologic features from Golestan 2017-2020.

It seems that the Golestan data set might benefit from further analysis. Multiple regression analysis would reveal the strength of associations between the various demographic and animal features and animal bites. Can you consult with a biostatistician on this matter?

TABLES

In Table 1, I think the factor Nationality can be deleted.

The title of Table 2 could be better: “Temporal aspects of the animal bites.”

Tables 2 and 3 have been switched. The text has Table 2 as the type of biting animal, but that information is shown in current Table 3.

In Table 3, the data on number and percent are switched from their proper headings.

RESULTS

Figures 1 and 2 are shown, but you say little about them. Why are they important? Can you elaborate a bit?

After Table 2, there is text on the completeness of treatment at the clinics. This seems extremely important to me. What is known about the outcomes of treatment? What can be done to improve the performance of the treatment services? Is the rabies service equally accessible in the different parts of Golestan? You do not show data on these points, and you do not discuss services.

DISCUSSION

I would delete the first sentence. You stated this in the Introduction.

A summary of your incidence data might be: “The rate of increase for animal bites in Golestan province parallels that in Iran as a whole, while the provincial incidence rate is substantially higher than the national rate.“

The final 2 words in the Discussion are out of place.

Reviewer #2: Dear author, in my point of view there are some points regarding your manuscript;

Would you please let me know that in cross sectional study design, can we calculate incidence?

In the method you mentioned that you used descriptive study such as mean and SD but I could not see it in the results.

In the table there are some values like 88/9, 1/1 which is not understandable.

It also need English editing

6. PLOS authors have the option to publish the peer review history of their article (what does this mean?). If published, this will include your full peer review and any attached files.

Reviewer #1: No

Reviewer #2: No

---

## [Author Response · Author response to Decision Letter 0]

11 Mar 2021

Dear Reviewers

We read your comments and made the requested corrections

- Details of the corrections are listed in the Response to Reviewers file.

- Referees' corrections are marked in highlighted form. (In Revised Manuscript with Track Changes file)

- Professional text editor corrections are also marked as Review Markup (In Revised Manuscript with Track Changes file)

- The letter of the professional editor certificate is also attached.

Thank you for your consideration of this manuscript.

Sincerely:

Irvan Masoudi Asl

---

## [Decision Letter · Decision Letter 1]

1 Apr 2021

PONE-D-20-38907R1

Trend and Epidemiological Patterns of Animal Bites in Golestan Province (Northern Iran) between 2017 and 2020

PLOS ONE

Dear Dr. masoudiasl,

Thank you for submitting your manuscript to PLOS ONE. After careful consideration, we feel that it has merit but does not fully meet PLOS ONE’s publication criteria as it currently stands. Therefore, we invite you to submit a revised version of the manuscript that addresses the points raised during the review process.

ACADEMIC EDITOR: Please address all comments made by reviewers . I am interested to see a detailed response to the comment made by reviewer #2.

We look forward to receiving your revised manuscript.

Kind regards,

Amir Radfar, MD,MPH,MSc,DHSc

Academic Editor

PLOS ONE

Journal Requirements:

Reviewers' comments:

Reviewer's Responses to Questions

**Comments to the Author**

1. If the authors have adequately addressed your comments raised in a previous round of review and you feel that this manuscript is now acceptable for publication, you may indicate that here to bypass the “Comments to the Author” section, enter your conflict of interest statement in the “Confidential to Editor” section, and submit your "Accept" recommendation.

Reviewer #1: All comments have been addressed

Reviewer #2: (No Response)

2. Is the manuscript technically sound, and do the data support the conclusions?

Reviewer #1: Yes

Reviewer #2: Partly

3. Has the statistical analysis been performed appropriately and rigorously? 

Reviewer #1: Yes

Reviewer #2: N/A

4. Have the authors made all data underlying the findings in their manuscript fully available?

Reviewer #1: No

Reviewer #2: Yes

5. Is the manuscript presented in an intelligible fashion and written in standard English?

Reviewer #1: No

Reviewer #2: Yes

6. Review Comments to the Author

Reviewer #1: Thank you for your revisions; the manuscript is much improved! I would like to suggest some further edits for clarity and concision. I also have one question about rabies vaccination and post-exposure prophylaxis.

ABSTRACT

Background : Rabies is one of the oldest zoonosis viral diseases, which still remains as one of the most important threats to public health in the 21st century.

Methods : This cross-sectional study examined epidemiologic features of all 33,996 cases of persons bitten by animals and referred to the rabies prophylaxis centers in Golestan province between March 2017 and March 2020. Factors included demographic information of the victim (age, gender, and occupation), type of invasive animals (dog, cat, and other types), time of bite (year, month, and hour), place of residence (urban or rural), and injury and treatment statuses. We also obtained national and provincial animal bite incidence data for all of Iran and for Golestan province for the longer interval 2013-2020 to examine broader time trends. We used SPSS version 19, QGIS version 3.1, and Excel 2013 to generate frequency distributions and descriptive statistics.

Results : The incidence rates of animal bites in Golestan province and Iran as a whole both increased smoothly. The latest incidence rate of animal bites in Golestan was 652 per 100,000 people, almost three times the overall national figure for 2020. Most cases of animal bites (67.6%) occurred in rural areas, and 36% of the victims aged under 19 years old. Dog and cat bites accounted for the great majority of cases (89% and 8%, respectively). The highest rate of animal bites was reported in the spring (30.8%). The lower limb was the most commonly bitten area in these individuals (64.6%). Of note, 87% of the cases received incomplete prophylactic post-exposure treatment, and 18% received immunoglobulin.

Conclusion : The increasing rates of animal bites in the study area as well as the higher rate compared to the national average indicates the need for further review of animal bite control programs.

INTRODUCTION

I would re-write the first sentence of the second paragraph thus: The virus is distributed worldwide with the exception of Antarctica. Then the next sentence can be deleted.

I think the first sentence of the final paragraph of the Introduction can be deleted.

METHODS

I would shorten the first paragraph:

We conducted a cross-sectional study including all 33,996 cases of persons bitten by animals and referred to anti-rabies centers for prophylaxis in Golestan province from March 2017 to March 2020. We also compiled national and Golestan data on animal bites reported from 2013-2020 to study longer-term incidence trends.

I would shift one sentence to make this the first sentence of the second paragraph:

Golestan provincial data reflecting epidemiological factors were collected from the electronic animal bites registration system.

I would re-write the sentence that currently begins with “As well, national data...” thus:

National data on overall and province-specific incidence rates 2013-2020 were collected from the Center for Communicable Diseases Control...

RESULTS

In the first paragraph, I would re-write the sentence that currently begins “Of note...”:

In Golestan province, Gomishan, Aliabad, and Maraveh-Tappeh counties had the highest incidence rates (934 to 957 cases per 100,000), while Minudasht county had the powest incidence (391 cases per 100,000) in 2019-2020 (see Figure 2).

I would make the names of the Figures more specific:

Fig.1. Incidence trend of animal bites per 100,000 people in Golestan province and in Iran

Fig. 2. Animal bite incidence rates in Golestan counties (2013-2020)

In the paragraph that describes the data in Table 1, if you add (Table 1) to an earlier sentence, you can delete the last sentence that currently reads “Some demographic information of the patients are shown in Table 1. That sentence does not really add much to the text.

In the text on Table 2, and in the Table itself, I think the hour intervals should be 0-6, 6-12, 12-18, and 18-24.

In the paragraph discussing Table 3, I think you should move the first two sentences should be moved to the end of the paragraph. That way, the structure of the text and of the Table are the same.

DISCUSSION

I would re-structure the first paragraph thus:

According to the World Health Organization’s report, Asia and Africa have the highest rate of animal bites [1]. In Iran, the rate of animal bites in most areas has an increasing trend [5]. We confirmed the trend up to the present: the incidence of animal bites in Iran has increased from 196 to 282 from 2013-2020. The rate of increase for animal bites in Golestan province paralleled that of Iran as a whole, although at substantially higher rates than the national rate. The current study showed that the rate of animal bites in Golestan has increased from 492 cases in 2013 up to 652 cases per 100.000 people in 2019-2020. One of the reasons for the higher rate of animal bites in this province may be due to the its location, which is in the center of animal husbandry and agriculture in Iran, naturally increasing the exposure of both humans and animals.

I would re-write some of the next paragraph:

Our data showed that the most animal bites in Golestan occurred in men. Animal bites have also been reported more commonly in men in other Iranian studies [8-13]. In some countries, the rate is higher among women [14], probably reflecting economic and behavioral differences. Unlike western countries, keeping animals at home is usually avoided in Iran. <additionally, activities="" and="" animals.="" be="" bites="" boldness="" can="" contact="" due="" high="" in="" men="" more="" non-occupational="" occupational="" of="" presence="" rate="" related="" society="" the="" their="" to="" with="">

<>I hope you can explain what you mean in the final sentence above; it is unclear to me.

Near the end of the Discussion, bottom of page 7, you write that only one case of human rabies resulted from this very large number of bites. You speculate that this likely resulted from the rather complete coverage of POST-exposure prophylaxis. But your own data in Table 3 shows that the great majority of cases had incomplete vaccine provided, and few received immunoglobulin. It seems very inconsistent to me. Perhaps all pet dogs and cats are vaccinated, thereby resulting in PRE-exposure prophylaxis? This is my big question, and I hope you can clarify it for me and other readers.

Finally, in the List of Tables and the List of Figures, please use the actual titles and not different wording.

Thank you for considering my suggestions!</additionally,>

Reviewer #2: Dear Author,

I could not find anything in your reference about calculation of incidence in cross sectional study. In my point of view we can not calculate incidence in cross sectional study but we can calculate prevalence. for calculation of incidence we have to conduct cohort study.

7. PLOS authors have the option to publish the peer review history of their article (what does this mean?). If published, this will include your full peer review and any attached files.

Reviewer #1: No

Reviewer #2: No

---

## [Author Response · Author response to Decision Letter 1]

11 Apr 2021

Dear Reviewers

We read your comments and made the requested corrections

- Details of the corrections are listed in the Response to Reviewers file.

- Referees' corrections are marked in highlighted form. (In Revised Manuscript )

Thank you for your consideration of this manuscript.

Sincerely:

Irvan Masoudi Asl

---

## [Decision Letter · Decision Letter 2]

29 Apr 2021

PONE-D-20-38907R2

Trend and Epidemiological Patterns of Animal Bites in Golestan Province (Northern Iran) between 2017 and 2020

PLOS ONE

Dear Dr. masoudiasl,

Thank you for submitting your manuscript to PLOS ONE. After careful consideration, we feel that it has merit but does not fully meet PLOS ONE’s publication criteria as it currently stands. Therefore, we invite you to submit a revised version of the manuscript that addresses the points raised during the review process.

ACADEMIC EDITOR: Please address all comments made by Reviewer #1

We look forward to receiving your revised manuscript.

Kind regards,

Amir Radfar, MD,MPH,MSc,DHSc

Academic Editor

PLOS ONE

Journal Requirements:

Reviewers' comments:

Reviewer's Responses to Questions

**Comments to the Author**

1. If the authors have adequately addressed your comments raised in a previous round of review and you feel that this manuscript is now acceptable for publication, you may indicate that here to bypass the “Comments to the Author” section, enter your conflict of interest statement in the “Confidential to Editor” section, and submit your "Accept" recommendation.

Reviewer #1: All comments have been addressed

Reviewer #2: All comments have been addressed

2. Is the manuscript technically sound, and do the data support the conclusions?

Reviewer #1: Yes

Reviewer #2: Yes

3. Has the statistical analysis been performed appropriately and rigorously? 

Reviewer #1: Yes

Reviewer #2: Yes

4. Have the authors made all data underlying the findings in their manuscript fully available?

Reviewer #1: Yes

Reviewer #2: Yes

5. Is the manuscript presented in an intelligible fashion and written in standard English?

Reviewer #1: Yes

Reviewer #2: Yes

6. Review Comments to the Author

Reviewer #1: The manuscript looks great! Thank you. May I suggest a few minor changes at this final stage?

In Table 2, it looks like the lowest bite rate was from midnight to 6AM (3%). I think you need to revise that sentence at the bottom of page 4.

In Table 2, can you change the label from ‘free’ to ‘freelancer’? That way, it will match the text in the Discussion.

At the top of page 5, the sentence could be made clearer if you write: “...96% of total bites; 49% of these animals had owners.”

In the middle of page 7 where you discuss the extraordinarily low death rate from rabies, the text appears inconsistent with the data on completeness of treatment. But the explanation you kindly provided in your response to reviewers taught me why there is no real inconsistency. (I am not a rabies expert!) I suggest that matters would be clarified in the article if you add some of the text from your response to reviewers here in the Discussion.

Reviewer #2: all of the comments are addressed .

7. PLOS authors have the option to publish the peer review history of their article (what does this mean?). If published, this will include your full peer review and any attached files.

Reviewer #1: No

Reviewer #2: No

---

## [Author Response · Author response to Decision Letter 2]

6 May 2021

Dear Reviewers

We read your comments and made the requested corrections

- Details of the corrections are listed in the Response to Reviewers file.

- Referees' corrections are marked in highlighted form. (In Revised Manuscript )

Thank you for your consideration of this manuscript.

Sincerely:

Irvan Masoudi Asl

---

## [Editor Report · Decision Letter 3]

10 May 2021

Trend and Epidemiological Patterns of Animal Bites in Golestan Province (Northern Iran) between 2017 and 2020

PONE-D-20-38907R3

Dear Dr. masoudiasl,

We’re pleased to inform you that your manuscript has been judged scientifically suitable for publication and will be formally accepted for publication once it meets all outstanding technical requirements.

Kind regards,

Amir Radfar, MD,MPH,MSc,DHSc

Academic Editor

PLOS ONE
---

## [Editor Report · Acceptance letter]

17 May 2021

PONE-D-20-38907R3 

Trend and Epidemiological Patterns of Animal Bites in Golestan Province (Northern Iran) between 2017 and 2020 

Dear Dr. Masoudi Asl:

I'm pleased to inform you that your manuscript has been deemed suitable for publication in PLOS ONE. Congratulations! Your manuscript is now with our production department. 

Kind regards, 

on behalf of

Dr. Amir Radfar 

Academic Editor

PLOS ONE